# Movement behaviours and physical, cognitive, and social-emotional development in preschool-aged children: Cross-sectional associations using compositional analyses

**Nicholas Kuzik[1], Patti-Jean Naylor[2], John C. Spence[1], Valerie Carson[1] ***

**1** Faculty of Kinesiology, Sport and Recreation, University of Alberta, Edmonton, Alberta, Canada, **2** Institute of Applied Physical Activity and Health Research, School of Exercise Science, Physical and Health Education, University of Victoria, Victoria, British Columbia, Canada

* vlcarson@ualberta.ca

**Data Availability Statement:** Data cannot be shared publicly because of ethical restrictions that only permit study team members to access the

## Abstract

### Background

Movement behaviours (e.g., sleep, sedentary behaviour, and physical activity) in isolation have demonstrated benefits to preschool-aged children's development. However, little is known on the integrated nature of movement behaviours and their relationship to healthy development in this age range. Thus, the objective of this study was to examine the relationships between accelerometer-derived movement behaviours and indicators of physical, cognitive, and social-emotional development using compositional analyses in a sample of preschool-aged children.

### Methods

Children (n = 95) were recruited in Edmonton, Canada. Movement behaviours were measured with ActiGraph wGT3X-BT accelerometers worn 24 hours/day. Physical (i.e., body mass index [BMI] z-scores, percent of adult height, and motor skills), cognitive (i.e., working memory, response inhibition, and vocabulary), and social-emotional (i.e., sociability, externalizing, internalizing, prosocial behaviour, and cognitive, emotional, and behavioural self-regulation) development were assessed. Objective height and weight were measured for BMI z-scores and percent of adult height, while the Test of Gross Motor Development-2 was used to assess motor skills. The Early Years Toolbox was used to assess all cognitive and social-emotional development indicators. Compositional linear regression models and compositional substitution models were conducted in R.

### Results

Children accumulated 11.1 hours of sleep, 6.1 hours of stationary time, 5.1 hours of light-intensity physical activity (LPA), and 1.8 hours of moderate- to vigorous-intensity physical activity (MVPA) per day. Movement behaviour compositions were significantly associated with physical (i.e., locomotor skills, object motor skills, and total motor skills) and cognitive (i.e., working memory and vocabulary) development ($R^2$ range: 0.11–0.18). In relation to

data. However, interested researchers may send requests for approval and data access to the University of Alberta Research Ethics Board (contact via reoffice@ualberta.ca).

**Funding:** The author(s) received no specific funding for this work.

**Competing interests:** The authors have declared that no competing interests exist.

other movement behaviours in the composition, MVPA was positively associated with most physical development outcomes; while stationary time had mixed findings for cognitive development outcomes (i.e., mainly positive associations in linear regressions but non-significant in substitution models). Most associations for LPA and sleep were non-significant.

## Conclusions

The overall composition of movement behaviors appeared important for development. Findings confirmed the importance of MVPA for physical development. Mixed findings between stationary time and cognitive development could indicate this sample engaged in both beneficial (e.g., reading) and detrimental (e.g., screen time) stationary time. However, further research is needed to determine the mechanisms for these relationships.

## Introduction

Sleep, sedentary behaviour, and physical activity—collectively referred to as movement behaviours—have received increased attention for their health benefits to preschool-aged children's development [1]. Systematic reviews of isolated movement behaviours have concluded more sleep, more physical activity, and less sedentary behaviour have numerous health benefits to aspects of physical, cognitive, and social-emotional development in preschool aged children [2–4]. However, considering that within a 24-hour period a change to one movement behaviour would necessitate compensation from another movement behaviour(s), the health benefits of movement behaviours in isolation may be misleading. For instance, if an intervention successfully increased a child's physical activity by 30 minutes in a day, then there would need to be 30 minutes less across the other movement behaviours. Thus, an integrated approach to understanding the health benefits of movement behaviours should be considered.

To date, little is known on the integrated nature of movement behaviours and their relation to healthy development in preschool-aged children [5]. In a recent systematic review of 10 studies examining combinations of movement behaviours, only physical development was examined and no studies included all movement behaviours [5]. Therefore, future research is needed on the collective relations between all movement behaviours with a broad range of developmental outcomes. Specifically, development can be categorized into three broad domains: physical (e.g., growth, motor skills, physical health), cognitive (e.g., executive functions, vocabulary), and social-emotional (e.g., emotional intelligence, relationship building) development [6]. However, to examine the collective relations between movement behaviours and these broad domains of development, methods that appropriately consider the codependent nature of movement behaviours are needed [5].

Individual movement behaviours are considered codependent because they cannot co-occur (mutually exclusive) and when all individual movement behaviours are summed they will equal the total time-frame sampled (exhaustive) [7]. Mutually exclusive and exhaustive properties of movement behaviours means this data is only meaningfully interpreted as a proportion of a whole, and thus are considered to have a constant sum constraint (values that always add to make a whole) [8]. One method that is capable of appropriately handling the codependent nature of movement behaviours is compositional analyses [7, 9]. Since the integrated movement behaviour systematic review [5], two studies have used compositional analyses to examine the associations between all movement behaviours and development outcomes

in preschool aged children [10, 11]. While health benefits were found for movement behaviours in both studies, only physical development outcomes were examined [10, 11]. Given the limited evidence, further research is needed to confirm previous findings on physical development as well as address the evidence gap related to cognitive and social-emotional development. Thus, the objective of this study is to examine the relations between accelerometer-derived movement behaviours and indicators of physical, cognitive, and social-emotional development using compositional analyses in a sample of preschool-aged children.

## Methods

### Participants and procedures

Data used in this analysis were collected as part of from the Parent-Child Movement Behaviours and Pre-School Children's Development study. Participants were children aged 3–5 years and their parents, whose primary language at home was English. Parents or guardians were recruited in Edmonton, Canada and surrounding areas through a local division of Sportball, a program that aims to teach children fundamental sport skills through play. Parents were approached in person by the lead investigator during Sportball summer camps and at Sportball classes. A total of 60/102 children were recruited from summer camps, but participation rates and reasons for non-participation from classes were not tracked due to logistical constraints. Additionally, the local Sportball organization distributed recruitment materials to parents via email and social media. It is unknown how many eligible parents received the email or viewed the social media posts, or their reasons for non-participation. In total, 131 parents or guardians agreed to participate. Ethical approval was obtained from the University of Alberta Research Ethics Board (Study ID: Pro00081175). Parents or guardians provided written informed consent

Data collection for this cross-sectional study occurred from July to November 2018. Children's gross motor development was measured at the University of Alberta. After the motor development assessment, parents and children were provided accelerometers, verbal and written study protocol instructions, and a log sheet to track sleep and accelerometer wear time. After the accelerometer wear period, the lead investigator visited the homes of parents or an alternative preferred location (n = 2) to collect the accelerometers. During the home visit, parents completed a questionnaire, which included the social-emotional development measures and socio-demographic measures, while children were administered cognitive development tasks. Additionally, children's height and weight were measured, and parents' height was also measured if they wanted assistance reporting their height in the questionnaire.

### Measures

**Movement behaviours.**   The children's movement behaviours included total sleep, stationary time (i.e., sedentary behaviour categorization in accelerometer data that contains no posture detection [12]), light-intensity physical activity (LPA), and moderate- to vigorous-intensity physical activity (MVPA). All movement behaviours were measured with ActiGraph wGT3X-BT accelerometers that were programmed at 30 Hz and given to a child and one parent. While 90–100 Hz is the recommended frequency for ActiGraph accelerometers in preschool-aged children, we chose 30 Hz to align with the validation studies that our movement behaviour cut-points are based on [13]. In nine cases, multiple preschool-aged children from the same family participated. Parents and children were instructed to wear the accelerometer on an elastic belt on their right hip for 24 hours a day over 7 days, except during water-based activities. Accelerometers were programmed to begin recording at the next instance of 00:00:00. When accelerometers were collected, data were downloaded in 15-second epochs for

both normal filter files and low frequency extension (LFE) filter files. Normal filtered files were used to categorize children's stationary time ($\leq$25 counts/15 seconds), LPA (26–419 counts/15 seconds), and MVPA ($\geq$420 counts/15 seconds), while LFE files were used to categorize total sleep [14]. While using shorter epochs may be advantageous to better represent the sporadic movement profiles of preschool-aged children, 15-second epochs were used to align with the validation studies that our movement behaviour cut-points are based on [13]. All movement behaviour categorization was conducted in R (version 3.6.1). For sleep, daytime (e.g., nap) and nighttime sleep were categorized through visual inspection guided by the log book, and heuristics according to previous visual inspection literature [15]. Sleep data was then merged with the normal filtered file, and non-wear time (i.e., >20 minutes consecutive 0 counts, no interruptions) was removed that was not sleep. Finally, days with <10 hours/day of waking day wear time were removed and participants with <3 days were removed.

**Physical development.** Physical development was operationalized as motor skills, adiposity, and growth. Motor skills were measured with the Test of Gross Motor Development– 2nd Edition (TGMD-2). Heights and weights were measured to calculate the surrogate adiposity measure of body mass index (BMI) z-scores. Growth was measured with heights, which were used to calculate child's percent of expected adult height.

The TGMD-2 assessed object skills, locomotor skills, and total motor skills. Testing consisted of six object motor skills (i.e., striking a stationary ball, dribbling, kicking, catching, overhand throwing, and underhand rolling) and six locomotor skills (i.e., running, galloping, hopping, leaping, horizontal jumping, and sliding) [16]. Children were divided into groups with one to five children in each group. Groups rotated around three to four stations that each had three to four skills and two different research team members. At each station, one team member took on the role of the facilitator while the other took on the role of the assessor. The facilitators main task was demonstrating and verbally explaining the skill two times for the children. Then each child was given one chance to practice the skill and two scored trials for each skill. The assessors main task was live scoring the children's attempts at performing the skill, as well as wearing a body camera that recorded a video of children's assessments to be scored later. All 12 skills were composed of three to five components, which were scored as demonstrated (i.e., 1) or not demonstrated (i.e., 0). Scores for both trials were summed across components to create an object motor skill score and a locomotor skill score, both out of a maximum 48 points. Object and locomotor skill scores were then summed to create a total motor development score. For each child, live scores coded by assessors and video scores coded by the lead investigator were compared for all pair-wise complete observations. Intraclass correlation coefficients (ICC; two-way, agreement) indicated moderate to good agreement for object motor (ICC = 0.719; 95% Confidence Interval (CI): 0.340, 0.860), locomotor (ICC = 0.693; 95% CI: 0.423, 0.825), and total motor skills (ICC = 0.791; 95% CI: 0.277, 0.915). Since live scores were scored by multiple assessors and video scores were scored by one assessor, video scored values were used for analysis. However, when a video score was missing, live scores were used for that observation. A recent systematic review of the TGMD-2 found several studies demonstrating moderate-strong criterion validity (e.g., r: 0.49–0.63 when compared to other motor development assessments), as well as excellent test-retest (ICC: 0.81–0.92), inter-rater (ICC: 0.88–0.93), and intra-rater reliability (ICC: 0.92–0.99) [17].

Children's height and weight were each measured twice with a stadiometer and digital scale, respectively. Children's weight was measured to the nearest 0.1 kg and height was measured to the nearest 0.1 cm. If a difference of $\geq$0.3 units were scored between the two measurements, a third measurement was performed and the average of the two closest measurements were used. Body mass index (BMI) z-scores were calculated according to the World Health Organization's (WHO) growth standards [18].

Children's height was measured with stadiometer as described above. The height of both biological parents was reported in the parental questionnaire. Parents also had the option to have their height measured with the stadiometer at the home visit so they could enter that value into the questionnaire. The child's current percent of expected adult height was calculated based on their current height and the average of their biological mother's and father's height, according to sex specific formulas [19].

**Cognitive development.** Response inhibition, visual-spatial working memory, and language development were employed as indicators of cognitive development. Based on pre-existing protocols [20–23], they were measured using the iPad-based Early Years Toolbox [24]. As parts of the toolbox, the Go/No-Go task was used to test response inhibition, the Mr. Ant task was used to test visual-spatial working memory, and the Expressive Vocabulary task was used to test language development. Visual and auditory instructions are built into each iPad task in order to standardize administration, however the lead investigator was also trained to provide further supplementary information when the child required clarification.

For the Go/No-Go task [20, 21], children were required to tap the screen when they saw a fish, which occurs 80% of the time (Go) but not tap the screen when they saw a shark, which occurs the remaining 20% of the time (No-Go). There were a total of three trials completed for all children with no changes in complexity. For each trial, 75 stimuli (fish or sharks) were presented in a semi-random order (i.e., no trial begins with a shark, and sharks are not presented consecutively more than twice) for 1,500 milliseconds followed by 1,000 milliseconds of no stimulus. Scores were calculated by multiplying the proportion of correct Go and No-Go stimuli (e.g., 160/180 correct Go stimuli multiplied by 30/45 correct No-Go stimuli = 0.593), with values closer to 1 indicating better response inhibition.

For the Mr. Ant task [22, 23], children saw Mr. Ant with sticker(s) (n = 1–8) on different parts of his body for 5 seconds, a blank screen for 4 seconds, and Mr. Ant again with auditory prompt to place stickers back on Mr. Ant. The task progressed in levels (n = 1–8 stickers) with three trials for each level to a maximum of 8 levels, and correspondingly a maximum of 8 points. The task ended after failure on all three trials within a level or successful completion of all eight levels. Starting at level 1, points were calculated as 1 point for each level with at least 2/3 trials correct. After a level was scored as 1/3 correct trials, that level and all subsequent levels were scored based on the number of correct trials, with 1/3 of a point for each correct trial.

For the Expressive Vocabulary task, children were presented with a maximum of 45 pictures and they were instructed to tell the lead investigator what the picture was. An incorrect description of the picture prompted the lead investigator to ask what else the item could be called, until the child correctly described the picture or until the lead investigator was confident that the child could not correctly produce the required word. Six incorrect descriptions in a row stopped the test, and points were calculated by summing the number of correct words.

The Early Years Toolbox has previously shown good to excellent reliability (Cronbach's α range: 0.84–0.95) for the internal consistency of response inhibition and expressive vocabulary, and moderate-strong criterion validity (r: 0.40–0.60) for the correlations between response inhibition, visual-spatial working memory, and expressive vocabulary with other validated tasks from the National Institute of Health's Toolbox and British Ability Scales [24]. In the present study, acceptable-good internal consistency reliability [25] was observed for go trials (Cronbach's α = 0.90), no-go trials (Cronbach's α = 0.78), and expressive vocabulary (Cronbach's α = 0.90).

**Social-emotional development.** Sociability, externalizing, internalizing, prosocial behaviour, and self-regulation (i.e., cognitive, emotional, and behavioural self-regulation) were the social-emotional development indicators used in this study. Social-emotional development

was measured using the paper-based Child Self-Regulation and Behaviour Questionnaire (CSBQ), which is also part of the Early Years Toolbox [24]. Parents completed 34-items, with responses ranging from 1 (not true) to 5 (certainly true). Subscales were calculated by averaging scores across items, while reverse scoring some items. Each subscale ranged from 0 to 5, with values closer to 5 being favourable for sociability, prosocial behaviour, and self-regulation, while values closer to 1 were favourable for internalizing and externalizing. When data was missing (n = 7), subscale averages were calculated without the missing items.

A previous study that used the first iteration of the questionnaire, with changes mainly consisting of going from 33 to 34 items in the current version, found that all subscales of the CSBQ had acceptable-good reliability (Cronbach's α range: 0.74–0.89) for internal consistency, and moderate-very strong correlations (r: 0.48–0.91) for analogous and nearest comparisons with Strengths and Difficulties Questionnaire subdomains [24]. In the present study, good internal consistency reliability [25] was observed for most subscales (Cronbach's α: 0.75–0.82), except for internalising (Cronbach's α = 0.55) and prosocial behaviour (Cronbach's α = 0.64).

**Covariates.** Based on previous movement behaviour and development research [26, 27], children's age, sex, ethnicity, number of siblings, and hours of childcare attendance, as well as parental age, relation to the child, education, income, marital status, type of home, and size of yard were considered as covariates. Child and parent age, on the day they received accelerometers, were calculated based on their date of birth reported on consent forms and questionnaires. Parent's were asked to select their "child's race/ethnicity (check all that apply)" from a list of 13 responses, and for analysis children were categorized as "White" or "underrepresented groups" due to the high prevalence of "Caucasian" responses, and heterogeneity across the other 12 possible response options. Number of siblings was scored ranging from "0" to "≥3" younger and older siblings, and classified as "0", "1", "≥2" total siblings. Childcare attendance was determined by asking parents in the questionnaire how many hours/week their child typically spends in care other than their own. Parental relationship to the child (i.e., "mother", "father", "other") was classified as "mother" or "father" since no one in this analytical sample selected "other". Seven response options for parental education ranged from "Less than high school diploma or its equivalent" to "University certificate, diploma, or degree above the bachelor's level". Parental income was based on 10 response options ranging from "Less than $25,000" to "More than $200,000" that increased by $25,000 at each choice, as well as a "Do not know" option. Two participants responded, "Do not know" and their responses were imputed to the sample median. Marital status was classified as "married" or "not married" because of the high prevalence of married responses and the heterogeneity across the other five possible response options. Home type was classified as "one level" or "two levels" based on nine possible response options, and an "other" response option where participants could specify their home type. Five response options for size of parent's yard ranged from "No yard at all" to "A large yard (eg ¼ acre block or larger)".

## Data analysis

Standard descriptive statistics were calculated for all outcome (physical = 5, cognitive = 3, social-emotional = 7) and demographic variables. Compositional descriptive statistics were calculated for the centrality and dispersion of movement behaviour data [28]. Centrality was defined by the closed geometric mean of all movement behaviours, normalized to 24-hours. Dispersion was calculated with a variation matrix that demonstrates the proportionality between two movement behaviours, with values closer to zero indicating a higher codependence.

Isometric log ratio transformations of the composition of movement behaviours (i.e., total sleep, stationary time, LPA, and MVPA) were calculated [28]. Regression models with only

movement behaviour composition variables and outcome variables were created to determine the overall influence of the composition of movement behaviours on each outcome variable. The coefficient of determination ($R^2$) indicated the effect size for the relation between movement behaviour compositions and the outcome variables. Next, simple linear regression models were conducted between each potential covariate and each outcome variable. Covariates were only included if they were significant in the simple linear regression models, such that each final model would only include covariates relevant to a particular outcome. Final models were then created for each outcome variable that included the pivot coordinates of isometric log ratio transformed movement behaviour compositions and covariates. The first pivot coordinate of each movement behaviour composition was considered to represent the influence of a single movement behaviour, in relation to the rest of the composition of movement behaviours, on each outcome variable.

Compositional substitution or time reallocation analyses were conducted according to methods proposed by Dumuid and colleagues [29]. Briefly, this analysis subtracts the predicted value of the outcome variable of the base regression model, from updated models that alter the movement behaviour composition variables according to a substitution of one movement behaviour for another movement behaviour. In total, 12 substitution models (e.g., reallocating 30 minutes of MVPA with 30 minutes of sleep) were created and compared to the base model, for each outcome variable. All substitutions looked at the change in outcome variables when 30 minutes of one movement behaviour was substituted for 30 minutes of another behaviour. To ensure that 30 minutes substitutions were plausible, the minimum amount of MVPA a participant accumulated (i.e., 47 minutes), as well as 1 standard deviation for time spent in MVPA (i.e., 28.8 minutes/day) were considered.

Assumptions for regression analyses (i.e., linearity, normality, and equal variance of residuals, as well as identifying influential observations) were checked through visual inspection of residuals (i.e., residuals vs fitted values, Q-Q, square root of Standardized residuals vs. fitted values, and Cook's Distance) and Shapiro-Wilk test of normality. Models with sociability, externalizing, internalizing, BMI, and total motor skills were significant in Shapiro-Wilk tests indicating multivariate non-normality. Transformations could not be completed for time reallocation models because they would disrupt the interpretation of results. Additionally, for other models, numerous transformations were applied to these outcomes and normality was not reached. Thus, participants were removed according to Cook's d values >4/n [30] and models were re-run as sensitivity analyses to determine if findings changed. All analyses were conducted in R (version 3.6.1) and statistical significance was set at $p < 0.05$.

## Results

From 131 participants, a total of 95 participants had usable accelerometer data and were included in the analysis (see Fig 1 for participant flow diagram). Aside from the analysis of response inhibition (n = 93; n = 2 software errors) and all motor skills outcomes (n = 93, n = 2 children chose not to participate), these 95 participants had data for all outcome variables. Children were predominantly boys (69.5%) with an average age of 4.5 years, and the average age for parents was 37.8 years (see Table 1 for participant characteristics). For the closed geometric mean of movement behaviours normalized to 24-hours, children accumulated 11.1 hours of sleep, 6.1 hours of stationary time, 5.1 hours of LPA, and 1.8 hours of MVPA. Additionally, the variation matrix values ranged from 0.15 (stationary time and MVPA), indicating the lowest co-dependence, to 0.02 (sleep and LPA), indicating the highest co-dependence between variables (see Table 2).

The composition of movement behaviours were significantly associated with three physical development outcomes (i.e., locomotor skills, object motor skills, and total motor skills) and

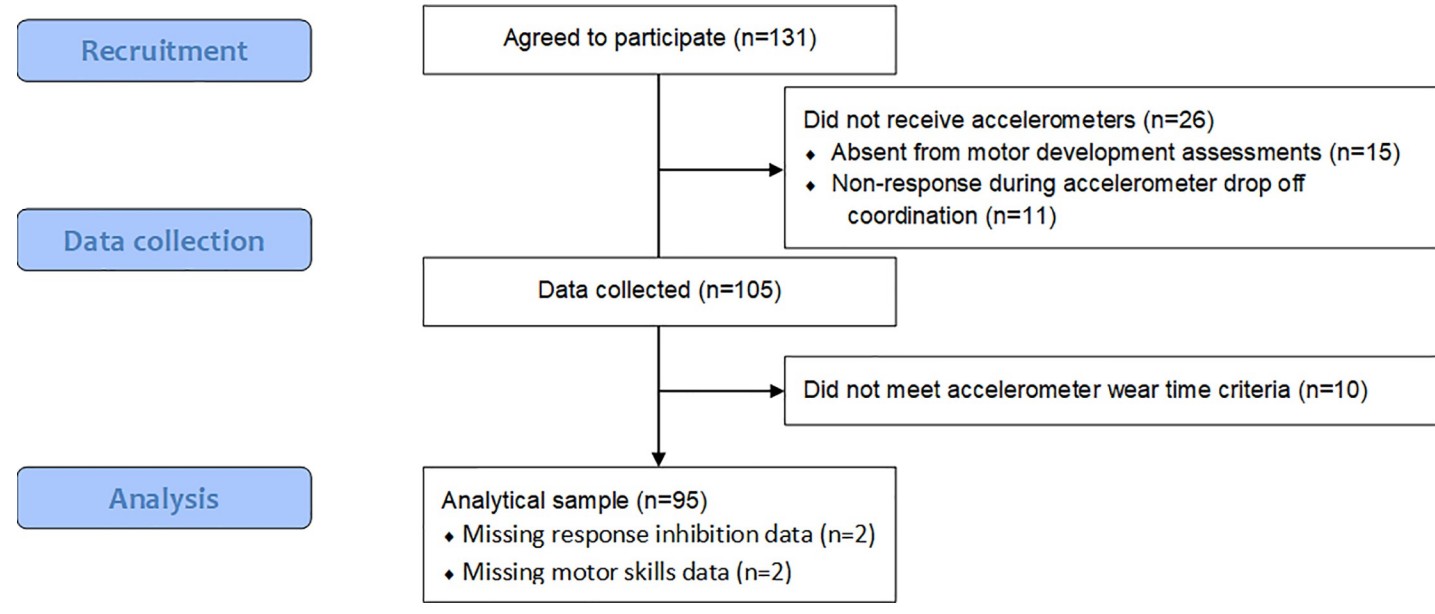

**Fig 1. Participant flow diagram.**

two cognitive development outcomes (i.e., working memory and vocabulary) (see Table 3). For all significant models, $R^2$ values were above 0.09 (Range: 0.11, 0.16) indicating medium effect sizes [31]. Covariates that were significantly associated across outcome variables and included in final regression models were: children's age, sex, ethnicity, number of siblings, as well as parental age, income, marital status, type of home, and size of yard (see Table 4 for all significant relations). Child's age was the most frequently included covariate in 7/15 of the final regression models, with parent's age and child sex being the next most frequently included with 3/15 models (see Table 4).

**Table 1. Outcome and covariate descriptive information.**

| Outcome Variable | Mean/Mode (SD/Percent) | Covariate Variable | Mean/Mode (SD/Percent) |
|---|---|---|---|
| Locomotor Skills | 27.8 (8.7) | Child Age (years) | 4.5 (0.7) |
| Object Motor Skills | 23.1 (7.1) | Sex | Male (69.5%) |
| Total Motor Skills | 50.9 (13.8) | Childcare (hours/week) | 21.2 (17.5) |
| BMI z-scores | 0.2 (0.9) | Ethnicity | Caucasian (71.6%) |
| Expected Adult Height (%) | 60.6 (3.8) | Siblings | One (54.7%) |
| Response Inhibition | 0.6 (0.2) | Parent Age (years) | 37.5 (5.1) |
| Working Memory | 1.9 (0.9) | Parent Education | Bachelor's degree (49.5%) |
| Vocabulary | 30.9 (7.2) | Parent Relation to Child | Mother (81.1%) |
| Behavioural Self-Regulation | 3.9 (0.7) | Marital Status | Married (89.5%) |
| Cognitive Self-Regulation | 3.7 (0.6) | Household Income | > $200,000 (25.3%) |
| Emotional Self-Regulation | 3.4 (0.8) | Home Type | Two levels (61.1%) |
| Externalizing | 2.1 (0.8) | Yard Size | Medium yard (69.5%) |
| Internalizing | 1.3 (0.4) | | |
| Sociability | 4.0 (0.7) | | |
| Prosocial Behaviour | 4.0 (0.6) | | |

BMI = Body mass index

**Table 2. Movement behaviour geometric mean (closed to 24 hours) and variation matrix.**

|  | LPA | MVPA | Sleep | Stationary |
|---|---|---|---|---|
| **Mean (hours/day)** | 5.09 | 1.75 | 11.12 | 6.05 |
| **LPA Variation** | 0 |  |  |  |
| **MVPA Variation** | 0.07 | 0 |  |  |
| **Sleep Variation** | 0.02 | 0.10 | 0 |  |
| **Stationary Variation** | 0.05 | 0.15 | 0.04 | 0 |

LPA = light-intensity physical activity; MVPA = moderate- to vigorous-intensity physical activity; Sleep = total sleep; Stationary = Stationary time. Values closer to zero indicate higher codependence.

Within compositional linear regression models, 5/20 significant relationships were found for physical development, 2/12 significant relationships were found for cognitive development, and 1/28 significant relationships were found for social-emotional development (see Table 5). For physical development, MVPA, relative to the other movement behaviours in the composition, was positively associated with object, locomotor, and total motor skills. While LPA, relative to the other movement behaviours in the composition, was negatively associated with object and total motor skills. For cognitive development, stationary time, relative to the other movement behaviours in the composition, was positively associated with response inhibition and vocabulary. For social-emotional development, MVPA, relative to the other movement behaviours in the composition, was positively associated with sociability. When removing multivariate influencers according to Cook's d, stationary time was significantly and negatively associated with BMI z-scores (n = 89), and MVPA was significantly and negatively associated with internalizing (n = 90).

Movement behaviour reallocations were associated with four outcome variables for physical development (i.e., BMI z-scores, object, locomotor, and total motor skills), one outcome

**Table 3. Outcome and movement behaviour composition full models.**

| Domain | Outcome Variable | $R^2$ | p value |
|---|---|---|---|
| **Physical[†]** | Locomotor Skills | 0.11 | 0.02* |
|  | Object Motor Skills | 0.18 | 0.00* |
|  | Total Motor Skills | 0.16 | 0.00* |
|  | BMI z-scores | 0.05 | 0.22 |
|  | Expected Adult Height (%) | 0.04 | 0.30 |
| **Cognitive[†]** | Response Inhibition | 0.08 | 0.07 |
|  | Working Memory | 0.11 | 0.01* |
|  | Vocabulary | 0.16 | 0.00* |
| **Social-Emotional** | Behavioural Self-Regulation | 0.00 | 0.98 |
|  | Cognitive Self-Regulation | 0.06 | 0.15 |
|  | Emotional Self-Regulation | 0.01 | 0.90 |
|  | Externalizing | 0.01 | 0.74 |
|  | Internalizing | 0.04 | 0.32 |
|  | Sociability | 0.08 | 0.05 |
|  | Prosocial Behaviour | 0.00 | 0.97 |

† = Movement behaviour compositions were significantly associated with the majority of outcome variables for the developmental domain (i.e., physical: 3/5; cognitive: 2/3; social-emotional: 0/7);

* = significant at $p < 0.05$

**Table 4. Significant outcome and covariate regression models.**

| Domain | Outcome | Covariate | Beta (p-value) |
|---|---|---|---|
| **Physical** | Locomotor Skills | Child Age (years) | 5.24 (0.00) |
| | Object Motor Skills | Child Age (years) | 3.58 (0.00) |
| | Total Motor Skills | Child Age (years) | 8.82 (0.00) |
| | BMI z-scores | Home Type (two levels) | -0.46 (0.01) |
| | Expected Adult Height (%) | Child Age (years) | 0.04 (0.00) |
| | | Sex (female) | 0.03 (0.00) |
| | | Parent Age (years) | 0.00 (0.04) |
| | | Household Income ($) | 0.01 (0.01) |
| **Cognitive** | Response Inhibition | Child Age (years) | 0.11 (0.00) |
| | | Sex (female) | 0.12 (0.01) |
| | Working Memory | Child Age (years) | 0.60 (0.00) |
| | Vocabulary | Child Age (years) | 6.79 (0.00) |
| | | Parent Age (years) | 0.33 (0.02) |
| | | Marital Status (not married) | -5.16 (0.03) |
| **Social-Emotional** | Cognitive Self-Regulation | Parent Age (years) | 0.03 (0.03) |
| | Emotional Self-Regulation | Siblings ($\geq$ 2) | -0.55 (0.03) |
| | Internalizing | Ethnicity (non-Caucasian) | -0.19 (0.04) |
| | Sociability | Yard Size (increasing size) | -0.26 (0.00) |
| | Prosocial Behaviour | Sex (female) | 0.26 (0.04) |
| | | Siblings ($\geq$ 2) | -0.41 (0.02) |
| | | Yard Size (increasing size) | -0.19 (0.01) |

Child age, parent age, household income, and yard size were treated as continuous variables and their unit is listed in parentheses; Home type, sex, marital status, siblings and ethnicity were treated as categorical variables and their comparator is listed in parentheses.

variable for cognitive development (i.e., vocabulary), and two outcome variables for social-emotional development (i.e., cognitive self-regulation and sociability) (see Table 6). For physical development, positive relationships were found when reallocating 30 minutes of another movement behaviour with 30 minutes of MVPA for BMI z-scores, object, locomotor, and total motor skills. Additionally, positive relationships were seen when reallocating LPA with stationary time for locomotor and total motor skills. For cognitive development, positive relationships were seen when reallocating sleep with stationary time for vocabulary. For social-emotional development, positive relationships were seen when reallocating another behaviour with MVPA for sociability and cognitive self-regulation. When removing multivariate influencers according to Cook's d, reallocating 30 minutes of MVPA with stationary time was significantly and positively associated with internalizing (n = 90).

## Discussion

The objective of this study was to examine the relations between accelerometer-derived movement behaviours and indicators of physical, cognitive, and social-emotional development using compositional analyses in a sample of preschool-aged children. Broad patterns for relations between movement behaviours and physical and cognitive development emerged across all analyses. However, associations with social-emotional development were less apparent. A summary of findings are presented in Tables 3 and 7.

For physical development, mainly motor development, a number of significant associations were observed for MVPA, relative to other movement behaviours, within linear regression and

**Table 5. Compositional linear regressions.**

| Outcome | LPA | MVPA | Sleep | Stationary |
|---|---|---|---|---|
| **Physical Development** | | | | |
| Locomotor Skills | -14.54 (0.07) | 9.05 (0.02)* | -3.80 (0.65) | 9.30 (0.10) |
| Object Motor Skills | -14.28 (0.02)* | 12.44 (0.00)* | 2.37 (0.72) | -0.54 (0.90) |
| Total Motor Skills | -28.82 (0.02)* | 21.49 (0.00)* | -1.43 (0.91) | 8.76 (0.29) |
| BMI z-scores | -1.07 (0.20) | 0.65 (0.11) | 1.07 (0.20) | -0.65 (0.24)⊖ |
| Expected Adult Height (%) | -0.02 (0.48) | 0.00 (0.79) | 0.02 (0.37) | -0.01 (0.59) |
| **Cognitive Development** | | | | |
| Response Inhibition | -0.10 (0.61) | 0.08 (0.43) | -0.26 (0.22) | 0.27 (0.047)* |
| Working Memory | 0.88 (0.24) | -0.33 (0.37) | -1.33 (0.10) | 0.78 (0.14) |
| Vocabulary | -4.44 (0.41) | 2.96 (0.25) | -8.56 (0.13) | 10.04 (0.01)* |
| **Social-Emotional Development** | | | | |
| Behavioural Self-Regulation | -0.10 (0.89) | -0.07 (0.84) | 0.24 (0.73) | -0.07 (0.88) |
| Cognitive Self-Regulation | -1.18 (0.05) | 0.52 (0.07) | 0.48 (0.42) | 0.17 (0.67) |
| Emotional Self-Regulation | 0.89 (0.28) | -0.14 (0.72) | -0.53 (0.51) | -0.21 (0.70) |
| Externalizing | -0.71 (0.36) | 0.37 (0.33) | -0.00 (1.00) | 0.34 (0.51) |
| Internalizing | -0.04 (0.92) | -0.20 (0.32)⊖ | 0.13 (0.75) | 0.11 (0.67) |
| Sociability | -0.64 (0.32) | 0.71 (0.02)* | -0.08 (0.91) | -0.00 (1.00) |
| Prosocial Behaviour | -0.50 (0.36) | 0.31 (0.26) | -0.22 (0.67) | 0.42 (0.26) |

LPA = light-intensity physical activity; MVPA = moderate- to vigorous-intensity physical activity; Sleep = total sleep; Stationary = Stationary time

* = significant at p < 0.05

⊕ = Became positively associated when removing influential participants according to Cook's d values >4/n

⊖ = Became negatively associated when removing influential participants according to Cook's d values >4/n

substitution models. However, relations for the other movement behaviours were predominantly null. For instance, reallocating 30 minutes of LPA with 30 minutes of MVPA resulted in higher locomotor and object motor skills by 3.28 and 3.99 units, which for a child aged 4.52 years (sample mean) would mean going from the 37th percentile to the 50th percentile of locomotor skills scores, and the 37th percentile to the 50th percentile (boys) or 63rd percentile (girls) of object motor skills [16]. This is line with a recent systematic review that found consistent positive relations between MVPA in isolation and motor development [2]. In contrast, LPA was negatively associated with motor skills in regression models and substitution models that reallocated stationary time with LPA. Future research is needed with tools that more accurately distinguish between sedentary behaviours and LPA in a larger more generalizable sample to better understand how these parts of the movement behaviour composition impact motor skills.

Beyond motor development, two other cross-sectional studies have used compositional analyses to examine the associations between movement behaviours and physical development in preschool children [10, 11]. For instance, the composition of movement behaviours was associated with BMI z-scores but not waist circumference [11]. Additionally, individual movement behaviours, relative to the other movement behaviours, did not demonstrate any significant relations. In another study, reallocating LPA and stationary time with sleep were all favourably associated with BMI z-scores at 3.5 years of age, while MVPA reallocations were not associated with BMI z-scores [10]. In contrast, findings from the current study suggest that reallocating stationary time with MVPA increased BMI z-scores by 0.2, and vice-versa. Previous research has shown that MVPA contributes to increased fat free mass and bone mass in preschool aged children [10, 32, 33], so the high volume of MVPA in this sample could be contributing to increased BMI z-scores through these mechanisms.

**Table 6. Significant substitution models (30 minutes).**

| Outcome | + Stationary—LPA | + Stationary—MVPA | + Stationary—Sleep | + LPA—Stationary | + LPA—MVPA | + MVPA—Stationary | + MVPA—LPA | + MVPA—Sleep | + Sleep—Stationary | + Sleep—MVPA |
|---|---|---|---|---|---|---|---|---|---|---|
| **Physical Development** | | | | | | | | | | |
| Locomotor Skills | 1.94 (0.26, 3.63) | NS | NS | -1.88 (-3.49, -0.26) | -3.82 (-6.93, -0.71) | NS | 3.28 (0.58, 5.97) | 2.12 (0.27, 3.98) | NS | -2.79 (-5.16, -0.42) |
| Object Motor Skills | NS | -3.67 (-5.35, -1.99) | NS | NS | -4.79 (-7.26, -2.32) | 2.75 (1.47, 4.04) | 3.99 (1.85, 6.14) | 2.62 (1.15, 4.09) | NS | -3.54 (-5.43, -1.66) |
| Total Motor Skills | 3.18 (0.65, 5.72) | -5.67 (-8.86, -2.49) | NS | -2.99 (-5.42, -0.57) | -8.62 (-13.30, -3.94) | 4.03 (1.60, 6.46) | 7.27 (3.20, 11.33) | 4.74 (1.96, 7.53) | NS | -6.33 (-9.90, -2.76) |
| BMI z-scores | NS | -0.23 (-0.46, -0.01) | NS | NS | NS | 0.19 (0.02, 0.36) | NS | NS | NS | NS |
| **Cognitive Development** | | | | | | | | | | |
| Vocabulary | NS | NS | 1.03 (0.18, 1.88) | -1.11 (-2.21, -0.01) | NS | NS | NS | NS | -1.08 (-1.95, -0.20) | NS |
| **Social-Emotional Development** | | | | | | | | | | |
| Cognitive Self-Regulation | NS | NS | NS | NS | -0.25 (-0.49, -0.01) | NS | 0.22 (0.01, 0.43) | NS | NS | NS |
| Internalizing | NS | NS$^{\oplus}$ | NS | NS | NS | NS$^{\ominus}$ | NS | NS$^{\ominus}$ | NS | NS$^{\oplus}$ |
| Sociability | NS | -0.21 (-0.39, -0.03) | NS | NS | -0.26 (-0.52, -0.00) | 0.16 (0.02, 0.29) | NS$^{\oplus}$ | 0.16 (0.01, 0.31) | NS | -0.21 (-0.40, -0.02) |

Stationary = Stationary time; LPA = light-intensity physical activity; MVPA = moderate- to vigorous-intensity physical activity; Sleep = total sleep; NS = non-significant

$\oplus$ = Became positively associated when removing influential participants according to Cook's d values >4/n

$\ominus$ = Became negatively associated when removing influential participants according to Cook's d values >4/n

For cognitive development, stationary time, relative to other movement behaviours, was associated with two out of three indicators of cognitive development in linear regression models. However, mainly null findings were observed for other movement behaviours in linear regression models. While three substitutions involving stationary time indicated it was

**Table 7. General Trends of significant relations.**

| Domain | Direction | LPA | | MVPA | | Sleep | | Stationary | |
|---|---|---|---|---|---|---|---|---|---|
| | | Linear | Substitution | Linear | Substitution | Linear | Substitution | Linear | Substitution |
| Physical | Favourable | 0 | 0 | **3** | **8** | 0 | 0 | 0 (+1) | 3 |
| | Unfavourable | 2 | 5 | 0 | 1 | 0 | 3 | 0 | 2 |
| | Null | **3** | **10** | 2 | 6 | **5** | **12** | 5 | **10** |
| Cognitive | Favourable | 0 | 0 | 0 | 0 | 0 | 0 | **2** | 1 |
| | Unfavourable | 0 | 1 | 0 | 0 | 0 | 1 | 0 | 0 |
| | Null | **3** | **9** | **3** | **9** | **3** | **8** | 1 | **8** |
| Social-Emotional | Favourable | 0 | 0 | 1 | 3 (+1) | 0 | 0 (+1) | 0 | 0 (+1) |
| | Unfavourable | 0 | 2 | 0 (+1) | 0 (+2) | 0 | 2 | 0 | 1 |
| | Null | **7** | **19** | **6 (-1)** | **18 (-3)** | **7** | **16 (-1)** | **7** | **17 (-1)** |

LPA = Light-intensity physical activity; MVPA = Moderate- to vigorous- intensity physical activity; Sleep = total sleep; Stationary = Stationary time; Numbers In parentheses' indicate number and direction of significant associations that were altered when removing influential participants according to Cook's d values >4/n; Bolded values indicate ≥50% associations were in that direction.

favourable for vocabulary scores, overall stationary time substitutions were predominantly null for cognitive development. Similarly, substitution models for other movement behaviours with cognitive development were all null. Since stationary time can only indicate low or no movement, and not what is qualitatively occurring during this time (e.g., screen time, time spent with parents reading, standing time), extrapolating the mechanism behind the favourable associations between stationary time and cognitive development in this sample is difficult. Previous systematic reviews that examined the health implications of sedentary behaviour in isolation found that parents reading with their children had beneficial associations with cognitive development, while screen time had unfavourable associations [4]. Therefore, one possible mechanism could be that children were engaging in more stationary time that was beneficial for cognitive development (e.g., reading) as opposed to stationary time that was unfavourable for cognitive development (e.g., screen time).

These results suggest that the composition of movement behaviours, measured with accelerometers, are important for some indicators of children's development. Determining the optimal levels in a 24-hour period of these behaviours is of high importance for public health recommendations. Similar to previous research using receiver operating characteristic curves to determine the ideal amount of MVPA, vigorous-intensity physical activity (VPA), and stationary time to distinguish between obese and non-obese children [34], future research could extend these findings and attempt to determine the optimal level of movement behaviours for healthy growth and development. However, in doing so, researchers should consider analyses sensitive to the compositional nature of all movement behaviours in a sample large enough to provide a wide spectrum of compositions.

Strengths of this study include the measurement of all movement behaviours via 24-hour wear time accelerometry, a broad array of developmental outcome measures, and the use of analyses sensitive to the compositional nature of movement behaviours. A limitation is the cross-sectional study design that prohibits understanding the causal mechanisms of the relationships observed. Additionally, the analytical sample was relatively small (n = 95) and only powered to detect medium-large effect sizes in models with <3 covariates, and large effect sizes in models with ≥3 covariates (i.e., percent of expected adult height, vocabulary, and pro-social behaviour). Lastly, convenience sampling from a physical activity program could have limited our generalizability. In fact, the average minutes/day of MVPA in this sample was 40 minutes higher compared to the national average, which could suggest poor generalizability to the broader population of Canadian preschool aged children [11].

In summary, this study used compositional analyses to examine the relations between movement behaviours across all domains of development (i.e., physical, cognitive, and social-emotional). The overall composition of movement behaviors appeared important for development. Broadly, MVPA was favourably associated with physical development, while mixed findings for stationary time indicated favourable or non-significant associations with cognitive development. Previous research has also demonstrated clear trends for favourable associations between MVPA and physical development—mainly motor development. Mixed findings between stationary time and cognitive development may indicate the inability of accelerometer research to distinguish between beneficial (e.g., reading) and detrimental (e.g., screen time) stationary time.

## Acknowledgments

The authors are grateful for all the children and parents. Sportball Edmonton for their tremendous support during recruitment and data collection. Additionally, we would like to thank Amanda Ebert, Anthony Bourque, April English, Autumn Nesdoly, Brendan Wohlers,

Carminda Lamboglia, Clara-Jane Blye, Evelyn Etruw, Jenna Davie, Kelsey Wright, Kevin Arkko, Madison Predy, Rebecca Rubliak, Ria Duddridge, Stephen Hunter, and Tyler Ekeli for their crucial assistance during the motor development data collection.

## Author Contributions

**Conceptualization:** Nicholas Kuzik, Patti-Jean Naylor, John C. Spence, Valerie Carson.

**Data curation:** Nicholas Kuzik.

**Formal analysis:** Nicholas Kuzik.

**Investigation:** Nicholas Kuzik.

**Methodology:** Nicholas Kuzik, Patti-Jean Naylor, John C. Spence, Valerie Carson.

**Project administration:** Nicholas Kuzik.

**Resources:** Valerie Carson.

**Software:** Nicholas Kuzik, Valerie Carson.

**Supervision:** Valerie Carson.

**Visualization:** Nicholas Kuzik.

**Writing – original draft:** Nicholas Kuzik.

**Writing – review & editing:** Nicholas Kuzik, Patti-Jean Naylor, John C. Spence, Valerie Carson.

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
