## [Decision Letter · Decision Letter 0]

10 Jun 2020

PONE-D-20-12402

Movement behaviours and physical, cognitive, and social-emotional development in preschool-aged children: cross-sectional associations using compositional analyses

PLOS ONE

Dear Dr. Carson,

Thank you for submitting your manuscript to PLOS ONE. After careful consideration, we feel that it has merit but does not fully meet PLOS ONE’s publication criteria as it currently stands. Therefore, we invite you to submit a revised version of the manuscript that addresses the points raised during the review process.

We look forward to receiving your revised manuscript.

Kind regards,

Javier Brazo-Sayavera, Ph.D.

Academic Editor

PLOS ONE

Journal Requirements:

3. We note you have included a table to which you do not refer in the text of your manuscript. Please ensure that you refer to Table 7 in your text; if accepted, production will need this reference to link the reader to the Table.

Remember that authors should make data fully available as it is stated below. 

**Comments to the Author**

1. Is the manuscript technically sound, and do the data support the conclusions?

Reviewer #1: Partly

Reviewer #2: Yes

2. Has the statistical analysis been performed appropriately and rigorously? 

Reviewer #1: Yes

Reviewer #2: Yes

3. Have the authors made all data underlying the findings in their manuscript fully available?

Reviewer #1: No

Reviewer #2: No

4. Is the manuscript presented in an intelligible fashion and written in standard English?

Reviewer #1: Yes

Reviewer #2: Yes

5. Review Comments to the Author

Reviewer #1: Movement behaviours and physical, cognitive, and social-emotional development in

preschool-aged children: cross-sectional associations using compositional analyses

The aim of this study was to examine the relationships between movement behaviours and indicators of physical, cognitive, and social emotional development in preschool-aged children. I strongly thing that the statistical approach has been perfectly chosen, furthermore, it also has a large number of analysed variables. Nevertheless, there are some issues that should be considered and which I will now discuss:

Methods

1. Participants and procedures, page 5, line 88. The sample was recruited through a “Sportball”, a program that aims to teach children fundamental sport skills through play, therefore we have a population bias. In this case, it was selected children and families who do physical activity regularly, moreover, the results showed that children had higher values of MVPA than those considered reference values.

2. Participants and procedures, page 6, line 100. Did the subjects have to complete all the tests to be included in the study? minimum percentage?

Measures

3. Movement behaviours, page 7, line 112. Did you consider the excessive counts?

Results

4. Page 15, line 297. The final sample was 95 participants, which was not very large. Did you calculate the statistical power?

5. Page 16. Table 2. The greatest co-dependence was shown between MVPA and stationary time, just as stationary time had a greater co-dependence with LPA against sleep. These values anticipate subsequent results in terms of motor skills, which may not be expected.

Particularly, the following results could be highlighted:

- LPA was significantly and negatively associated with object and total motor skills. Moreover, locomotor and total motor skills also showed composition full models that included the stationary time as a positive component.

- In relation to the substitution models, both in locomotor and total motor skills, to replace LPA for stationary time showed a significant and positive effect; as well as, the opposite replacement was significantly negative. Furthermore, to replace LPA for MVPA showed a significant and positive effect on locomotor skills, but no significant results were found in the replacement of stationary time for LPA.

Consequently, these results suggest that LPA could be more detrimental in terms of motor skills than the stationary time. How could this be explained?? I This important question has not been considered in the discussion section.

Reviewer #2: This paper explores the 24-hour integrated movement behaviours and their association with health outcomes in pre-school aged children. The authors explore outcomes in the domains of: physical, cognitive and social-emotional development. The authors state that physical development has been the primary focus of previous studies in the field, and a limitation of previous studies is that there is little to no consideration for all behaviours over the 24-hour day. This study, therefore, addresses two gaps by exploring movement behaviours as a complete 24-hour profile as well as exploring outcomes in other domains of development. The methods or rigorous and sound, including the statistical analyses. This paper adds to the growing body of literature around the importance of the 24-hour profile of movement behaviours. My specific comments are below.

Comments, Major:

So that the results are easier to follow, I suggest the results are presented in reference to the outcome domains of Physical development, Cognitive development, Social-emotional development (that is, in the text as well as in the Tables). There are a lot of outcomes and it is hard to follow at times. The authors stated in the introduction that a strength of this paper was that associations between movement behaviours and other non-physical development outcomes were explored, so I think this needs to be reflected in how the results are presented. This is in fact how the Discussion is laid out.

The language around the time reallocations is at time confusing. Consider using language that better reflects what was actually done, i.e., reallocation or replacing time. Adding/subtracting is technically correct, but it is more true to say that the time is reallocated. E.g., line 348 of Discussion “For instance, adding 30 minutes of MVPA while subtracting 30 minutes of LPA resulted in higher locomotor and object motor skills by 3.28 and 3.99 units” would be clearer as “replacing/reallocating 30 min of LPA with 30-min of MVPA”.

Overall, I think the results need to be written clearer. It is hard to follow what the authors are trying to say, and what are the main messages they want the reader to take away. There are a lot of outcomes and a lot of analyses with the entire 24-hour composition and with the reallocations. All good work, but needs to be clearer.

The authors need to consider and mention the implications of the cross-sectional design in the Discussion.

Comments, Minor

Is BMI really a measure of adiposity? I suggest not. Please consider wording around this and changing to something like body size.

It is not entirely clear where participants were recruited from. This sentence is not clear “Parents or guardians were recruited in Edmonton, Canada and surrounding areas through a local division of Sportball, a program that aims to teach children fundamental sport skills through play, as part of the Parent-Child Movement Behaviours and Pre-School Children’s Development study”. Were they recruited through Sportball, which was part of this other study? Please clarify.

Line 113: close bracket missing

How did the sample go from 131 to 95? Why were data missing/invalid?

What were the movement behaviour volumes in this cohort compared to the Canadian movement behaviour guidelines for this age group? Would be helpful information for the reader up front in the results. As mentioned in the limitations section.

The higher MVPA of this cohort could be a result of the sampling method. That is, participants were recruited from the Sport Ball program. Please acknowledge this more clearly in methodological considerations.

For Table 1, it would be helpful to see the range of possible scores and the direction (what high/low scores represent) for the outcomes.

Table 1, consider use of decimals. For example, does age really need to be to two decimals (same as in Discussion).

Table 2 add a footnote stating that closer to 0=greater codependence.

Table 3, indicate clearly in the title that these were adjusted for covariates.

6. PLOS authors have the option to publish the peer review history of their article (what does this mean?). If published, this will include your full peer review and any attached files.

Reviewer #1: No

Reviewer #2: No

---

## [Author Response · Author response to Decision Letter 0]

6 Jul 2020

Please see uploaded response to Reviewers document.

---

## [Decision Letter · Decision Letter 1]

20 Jul 2020

PONE-D-20-12402R1

Movement behaviours and physical, cognitive, and social-emotional development in preschool-aged children: cross-sectional associations using compositional analyses

PLOS ONE

Dear Dr. Carson,

Thank you for submitting your manuscript to PLOS ONE. After careful consideration, we feel that it has merit but does not fully meet PLOS ONE’s publication criteria as it currently stands. Therefore, we invite you to submit a revised version of the manuscript that addresses the points raised during the review process.

Probably due to the increase in the academic duties along the special period that world is living right now, availability of reviewers is compromised. Then, I respect authors' time and implication in the peer-review process and I have added some comments for reflecting. As it has been mentioned by reviewers and by myself, the work is interesting for the scientific field. However, I am sure that authors would like to publish a document without mistakes or missunderstoods. Please, take in this sense the considerations. 

We look forward to receiving your revised manuscript.

Kind regards,

Javier Brazo-Sayavera, Ph.D.

Academic Editor

PLOS ONE

Additional Editor Comments (if provided):

I congratulate the authors becase the current study add relevant insights to the scientific literature in this field.

Two reviewers completed their reviews in a first round but for the second review only one of them was available. However, I have checked out reviewer's 2 comments and I think there are minor issues still to address (pages and lines are referred to the tracked document):

P7 L116-133: Due to the relevance of accelerometry for this study, I think it is important to report the software you used to calculate variables that you used later for analyses.

In addition, I do not know why authors selected 30 Hz as sampling frequency when the recommendation for this age group is 90-100 Hz. Also, I understand that 15s epochs are enough, but considering the quick changes in this age group, shorter epochs could provide more confidence to the results.

P14 L287-290: Following reviewer’s 2 recommendations and after reading the sentence you have added respect terms “adding or subtracting”, I encourage to reconsider using terminology that reviewer’s 2 recommend, which I consider more appropriate for that case.

P14 L 294: Minutes should be in plural.

P15 L310-315: Please, consider to create a flow chart or a figure that could explain easier this flow of participants that you explain at the beginning of the results section.

P15 L316: Please, remove a dot after “variables”. It is duplicated.

P18 L330-414: I understand that it is difficult to place tables in the correct point, but I think you have to consider to move tables closer to the text that cites them in order to do it clearer. You cite tables 3, 4 and 5 that are in other parts of the text. Consider that there are a lot of outcomes, all necessaries, but it is important to do it clearer for the reader.

Reviewer's Responses to Questions

**Comments to the Author**

1. If the authors have adequately addressed your comments raised in a previous round of review and you feel that this manuscript is now acceptable for publication, you may indicate that here to bypass the “Comments to the Author” section, enter your conflict of interest statement in the “Confidential to Editor” section, and submit your "Accept" recommendation.

Reviewer #1: All comments have been addressed

2. Is the manuscript technically sound, and do the data support the conclusions?

Reviewer #1: Yes

3. Has the statistical analysis been performed appropriately and rigorously? 

Reviewer #1: Yes

4. Have the authors made all data underlying the findings in their manuscript fully available?

Reviewer #1: (No Response)

5. Is the manuscript presented in an intelligible fashion and written in standard English?

Reviewer #1: Yes

6. Review Comments to the Author

Reviewer #1: The authors have changed the manuscript and included all my suggestions. It is true that the statistical power is not very high, however the statistics are totally correct and it is definitely an interesting article

7. PLOS authors have the option to publish the peer review history of their article (what does this mean?). If published, this will include your full peer review and any attached files.

Reviewer #1: No

---

## [Author Response · Author response to Decision Letter 1]

3 Aug 2020

Please see attached response to reviewers file.

---

## [Editor Report · Decision Letter 2]

6 Aug 2020

Movement behaviours and physical, cognitive, and social-emotional development in preschool-aged children: cross-sectional associations using compositional analyses

PONE-D-20-12402R2

Dear Dr. Carson,

We’re pleased to inform you that your manuscript has been judged scientifically suitable for publication and will be formally accepted for publication once it meets all outstanding technical requirements. Congratulations for the study. 

Kind regards,

Javier Brazo-Sayavera, Ph.D.

Academic Editor

PLOS ONE

---

## [Editor Report · Acceptance letter]

7 Aug 2020

PONE-D-20-12402R2 

Movement behaviours and physical, cognitive, and social-emotional development in preschool-aged children: cross-sectional associations using compositional analyses 

Dear Dr. Carson:

I'm pleased to inform you that your manuscript has been deemed suitable for publication in PLOS ONE. Congratulations! Your manuscript is now with our production department. 

Kind regards, 

on behalf of

Dr. Javier Brazo-Sayavera 

Academic Editor

PLOS ONE